# The Mystery of Extramitochondrial Proteins Lysine Succinylation

**DOI:** 10.3390/ijms22116085

**Published:** 2021-06-04

**Authors:** Christos Chinopoulos

**Affiliations:** Department of Biochemistry and Molecular Biology, Semmelweis University, 1094 Budapest, Hungary; chinopoulos.christos@eok.sote.hu; Tel.: +36-1-459-1500 (ext. 60024); Fax: +36-1-267-0031

**Keywords:** succinyl-CoA, ketoglutarate dehydrogenase complex, post-translational modification, lysine, peroxisomes, fatty acid oxidation

## Abstract

Lysine succinylation is a post-translational modification which alters protein function in both physiological and pathological processes. Mindful that it requires succinyl-CoA, a metabolite formed within the mitochondrial matrix that cannot permeate the inner mitochondrial membrane, the question arises as to how there can be succinylation of proteins outside mitochondria. The present mini-review examines pathways participating in peroxisomal fatty acid oxidation that lead to succinyl-CoA production, potentially supporting succinylation of extramitochondrial proteins. Furthermore, the influence of the mitochondrial status on cytosolic NAD^+^ availability affecting the activity of cytosolic SIRT5 iso1 and iso4—in turn regulating cytosolic protein lysine succinylations—is presented. Finally, the discovery that glia in the adult human brain lack subunits of both alpha-ketoglutarate dehydrogenase complex and succinate-CoA ligase—thus being unable to produce succinyl-CoA in the matrix—and yet exhibit robust pancellular lysine succinylation, is highlighted.

## 1. Background

Succinylation of lysine residues is a post-translational modification occurring in hundreds of proteins which can happen enzymatically or non-enzymatically [1,2,3,4,5,6,7]. In either case, the high-energy metabolite, succinyl-CoA, is required. It is a common misconception that succinyl-CoA is exclusively produced inside the mitochondrial matrix by either (i) α-ketoglutarate dehydrogenase complex (KGDHC) (ii) the reversal of succinate-CoA ligase (SUCL); (iii) the catabolism of valine, isoleucine, methionine, thymine, and odd-number chain fatty acids (and perhaps propionate); or iv) through the reaction catalyzed by 3-oxoacid CoA-transferase 1 (OXCT1), see Figure 1. In fact, succinyl-CoA is also produced within peroxisomes—see Section 4—and there are potential routes through which the peroxisomally-generated succinyl-CoA finds its way to the cytosol (discussed below), unlike mitochondrially-made succinyl-CoA that is trapped in the matrix. The present mini-review focuses exclusively on delineating the extramitochondrial pathways yielding succinyl-CoA; furthermore, since some of those pathways depend on NAD^+^ availability, the connection of cytosolic protein lysines succinylation to mitochondrial status is examined. Finally, the notion that glial cells in the adult human brain are known to lack KGDHC and succinate-CoA ligase subunits in their mitochondria, yet exhibit robust pancellular protein lysine succinylation, is emphasized. For excellent reviews regarding the role and regulation of this post-translational modification, the reader is referred to [8,9,10,11,12].

## 2. Lysine Succinylation Inside and Outside Mitochondria

Protein lysine succinylation is more prevalent inside than outside mitochondria due to the high-rate generation of succinyl-CoA in the citric acid cycle, and the much greater abundance of coenzyme A (CoASH) and acyl-CoAs in the matrix compared to the cytosol [13,14,15]. Although succinyl-CoA production has been originally attributed to succinate-CoA ligase reversal, we now know that it is rather due to KGDHC activity during glutamate catabolism. This was recently demonstrated in erythropoiesis [16] and is likely to be the case during most other pathophysiological conditions [17,18,19]. Indeed, succinate-CoA ligase deficiency leads to hyper-succinylation of proteins [20]. Nonetheless, it is firmly established that lysine succinylation also occurs outside mitochondria [2,20,21,22,23]. Furthermore, carnitine palmitoyltransferase (CPT), an enzyme attached to the outer mitochondrial membrane facing the cytosol, exhibits lysine succinyltransferase activity [5]. However, mindful of the impermeability of succinyl-CoA across membranes, and with no known transfer mechanism—unlike other acyl-CoAs [24,25]—the question arises as to how this metabolite gets to the cytosol, and from which pathway(s).

## 3. Extramitochondrial Sites of Succinyl-CoA Production

According to *Metabolic Atlas* (https://www.metabolicatlas.org/ (accessed on 20 May 2021)) [26], succinyl-CoA can emerge in the cytosol (including the nucleus) through any of the following three routes, see also Figure 1:

(1) Succinylcarnitine + CoA ⇔ carnitine + succinyl-CoA, mediated by carnitine palmitoyltransferase, [27]. The enzyme exhibits three isoforms with tissue-specific distribution: 1A (liver), B (muscle) or C (brain), all located in the outer mitochondrial membrane [28,29], while the C isoform also present in ER [30]. CoA is available in the cytosol by many routes, while succinylcarnitine can be transported from peroxisomes through the carrier SLC25A20. Originally reported as a “mitochondrial carnitine/acylcarnitine carrier protein”, it is now known to reside in peroxisomes as well, see Section 4. Succinylcarnitine can be made in peroxisomes as follows: carnitine + succinyl-CoA ⇔ succinylcarnitine + CoA through carnitine O-octanoyltransferase (CROT) [31,32], reviewed in [33]. In human peroxisomes, the carnitine required for this reaction may be imported by one of the Organic Cation Transporters (OCTs). Rodent peroxisomes express the OCTN3 (Organic Cation Transporter 3) (SLC22A21) carnitine transporter [34,35]. However, a peroxisomal localization of OCTN3 is not being universally accepted [36]. The multiple routes of succinyl-CoA formation within peroxisomes is addressed in Section 4.

(2) Transport of succinyl-CoA from mitochondria to the cytosol. This is based on a model assumption published in [37] attempting to generate a comprehensive, high-quality genome-scale metabolic reconstruction by Thiele and Palsson, but has never been experimentally verified;

(3) ATP + CoA + succinate ⇒ AMP + PPi + succinyl-CoA, potentially mediated by a microsomal dicarboxylyl-CoA synthetase (E.C. 6.2.1.23). In rat liver and kidney (but not muscle), dicarboxylic acids (succinate is a dicarboxylic acid) can be converted into their CoA esters by such an enzyme [38,39]. However, it has also been suggested that the enzyme is not active on short-chain dicarboxylic acids such as succinate, because enzyme activity leaned toward zero by decreasing carbon chain length of the dicarboxylic acid, and with C = 5 it was already minimal [38].

In conclusion, it is the author’s opinion that the only viable route of succinyl-CoA provision to the cytosol is through formation of succinyl-CoA within peroxisomes, followed by its export in the form of succinylcarnitine. Thus, the question arises: how does succinyl-CoA form within peroxisomes?

## 4. Intraperoxisomal Routes of Succinyl-CoA Production

Evidence for the presence of succinyl-CoA within peroxisomes has been circumstantial for decades, stemming from at least two findings: (i) presence of dicarboxylic acids (including succinate) and 3-oxoadipate (3-ketoadipate) in urine which could be precursors to succinyl-CoA in peroxisomes [40,41,42]; (ii) expression of ACOT4 (an acyl-CoA thioesterase) in peroxisomes, an enzyme capable of hydrolyzing specifically succinyl-CoA [41,43]. Regarding (i), the presence of dicarboxylic acids in urine was mentioned as early as 1934 [44,45,46,47,48]. Dicarboxylic acids are formed by ω oxidation of monocarboxylic fatty acids. This occurs by an initial hydroxylation of the ω- or the ω-1 carbon by microsomal CYP4A enzymes [49], followed by further oxidations by alcohol and aldehyde dehydrogenases yielding the dicarboxylic acid [50], a process collectively known as ω-oxidation [51]. On the other hand, β-oxidation of very long dicarboxylic fatty acids occurs in peroxisomes [27]. Dicarboxylic acids may enter peroxisomes as the CoA ester, followed by chain shortening through peroxisomal β-oxidation. The individual reactions of the peroxisomal β-oxidation are catalyzed by different enzymes encoded by distinct genes compared with the mitochondrial β-oxidation system [52]. Overall, the dicarboxylic acids oxidized in peroxisomes may originate from mitochondrially produced monocarboxylic acids [48,53,54] and/or also very long-chain and long-chain monocarboxylic fatty acids through ω-oxidation [55,56]. Very long-chain and long-chain monocarboxylic fatty acids are mostly of plant origin. Current consensus though is that most of the carboxylic acids are activated to the corresponding CoA ester outside the peroxisome by acyl-CoA synthetases localized in the peroxisomal membrane or in the ER [57]. These acyl-CoA esters are then transported across the peroxisomal membrane via a family of ABC transporters. At least four peroxisomal ABC half-transporters have been identified, as reviewed in [58,59]. Regarding adipic acid, it is found in beets but is also used as a food additive (code named: E355) yielding succinyl-CoA in peroxisomes through adipoyl-CoA, see Figure 1. Finally, regarding 3-ketoadipate (3-oxoadipate), although its presence in human urine is well documented [60], the pathway forming it is not yet known. 3-ketoadipate may lead to succinyl-CoA formation in peroxisomes through the concerted action of fatty acid oxidation and peroxisomal 3-ketoacyl-CoA thiolase (ACAA1), see Figure 1.

Regarding (ii), nine functional acyl-CoA thioesterases and acyltransferases have been identified in mice and four in humans [61,62,63,64,65,66]. Human peroxisomes harbor two acyl-CoA thioesterases, ACOT4 and ACOT8, while mice express four (ACOT3-6) [43,67]. Acyl-CoA thioesterases hydrolyze acyl-CoAs to the free fatty acid and CoASH. ACOT4 exhibits selectivity towards succinyl-CoA (Km ≈ 13 μM) and glutaryl-CoA (Km ≈ 37 μM) [41], catalyzing the reaction H_2_O + succinyl-CoA -> CoA + H^+^ + succinate. Acyl-CoA thioesterase activity was first identified in the 1950s, where a succinyl-CoA thioesterase (most likely ACOT4) was partially purified from pig heart [68]. The existence of ACOT4 in peroxisomes strongly suggests that succinyl-CoA is formed within peroxisomes.

From the above considerations, it is inferred that succinyl-CoA is formed within peroxisomes from long and very long fatty acids (mostly from plant origin), 3-oxoadipate, and adipic acid through multiple routes. Succinyl-CoA is then transformed to succinylcarnitine (through CROT), and now the need arises to export succinylcarnitine through the peroxisomal membrane. For this, the most likely candidate is the SLC25A20 carrier. There is strong evidence of expression of SLC25A20 in peroxisomal membranes, but this is not unequivocally proven [31,32,69,70]. However, peroxisomes do not harbor metabolic pathways using acetyl-CoA, one of the major end products of fatty acid oxidation; therefore, they rely on mechanisms for exporting such products [59]. The SLC25A20 carrier may fill this niche, and thus become the means for succinylcarnitine export to the cytosol.

## 5. What Is the Connection between the Mitochondrial Status and Extramitochondrial Proteins Lysine Succinylation?

In [71], the conclusion was reached that the extent of proteins lysine succinylation varies with the metabolic state but does not correlate with the mitochondrial NAD^+^/NADH ratio. Their experiments unequivocally demonstrated that inhibition of the citric acid cycle or the electron transport chain; the use of uncouplers; or the altering of oxygen partial pressure—all known to impact the NAD^+^/NADH ratio but in a different manner—exerted various effects on proteins lysine succinylation, hinting at no pattern. However, in that study, they specifically examined the succinylation of lysines in mitochondrial proteins. On the other hand, in [20], the authors showed that mutations in the gene coding of a critical subunit of succinate-CoA ligase causes global protein succinylation, not just those found in mitochondria. The authors postulated that this could be due to secondary deficiency of sirtuin SIRT5 attributed to the depletion of NAD^+^ levels, similarly described in other diseases encompassing mitochondrial dysfunction [72,73,74]. In the author’s opinion, the cytosolic isoforms iso1 and/or iso4 of SIRT5 mediate the connection between the mitochondrial metabolic state and cytosolic proteins lysine succinylation. To elaborate on this further: Sirtuins are a group of evolutionary-conserved proteins named after a yeast gene called “Silent mating-type Information Regulation 2” [75]. Mammalian cells express seven sirtuin proteins, denoted as SIRT1-SIRT7 [76]. Sirtuins exhibit partially overlapping, but distinct deacylase activities [77]. SIRT1, SIRT2, and SIRT3 are deacetylases; SIRT4, which was originally demonstrated to be an ADP-ribosyltransferase and lipoamidase [78] is a demethylglutarylase, dehydroxymethylglutarylase, and demethylglutaconylase [79]; SIRT5 is a robust desuccinylase [80]; SIRT6 has weak deacetylase activity, but more efficient deacylase activity towards long chain fatty acyl groups such as myristoyl- and palmitoryl-groups [77]; SIRT7 also shows desuccinylase activity but due to its location it only acts on histones. It has additionally been shown to act as a long-chain deacylase [81]. Sirtuins not only exhibit distinct deacylase activities, but also different subcellular distribution [82]. SIRT1 and SIRT2 reside predominantly in the cytoplasm and can translocate into the nucleus under certain conditions [83]. SIRT3, SIRT4, and SIRT5 were originally considered as predominantly mitochondrial proteins [82,84,85,86] but, more recently, SIRT5 isoforms iso1 and iso4 were reported to reside in the cytoplasm [87,88]. SIRT6 and SIRT7 reside in the nucleus [89] and nucleolus [82], respectively.

The localization of SIRT5 iso1 and iso4 in the cytosol may be key to understanding why the mitochondrial state influences the extent of cytosolic proteins lysine succinylation. SIRT5 (and SIRT7) removes succinyl adducts from lysines in the presence of NAD^+^ [80,90]; the reaction catalyzed is:

H_2_O + N6-succinyl-L-lysyl-[protein] + NAD^+^ ->-O-succinyl-ADP-D-ribose + L-lysyl-[protein] + nicotinamide.

From the reaction shown above, it is immediately apparent that lack of NAD^+^ would lead to lack of desuccinylation, and thus protein lysines would remain succinylated. Furthermore, how is the cytosolic NADH/NAD^+^ ratio regulated, and what is the connection with mitochondria? In eukaryotic cells, the mitochondrial NAD/NADH ratio is approximately 5–10, while in the cytoplasm it ranges between 100 and 1000 [91,92,93]. While separate, the cytoplasmic and mitochondrial NAD^+^+NADH pools are connected by either (i) biochemical pathways relying on NAD^+^/NADH and/or (ii) NAD^+^ biosynthesis. In mammals, NAD^+^ biosynthesis may occur via four different routes: de novo synthesis from tryptophan, synthesis from vitamin B_3_, nicotinamide or nicotinic acid, or conversion of nicotinamide riboside [94]. By definition, NAD^+^ biosynthetic pathways take much longer than alterations in NAD^+^/NADH ratios [95]; consequently, they are less likely to play a role in the short run and, thus, are not subject to metabolic alterations as they may rapidly occur. Therefore, in all likelihood, reactions in which NADH/NAD^+^ are substrates or products influence SIRT5 iso1 and iso4 activity and are the potential culprits for affecting cytosolic proteins lysine succinylation. According to *Metabolic Atlas*, NADH/NAD^+^ participate in >200 reactions in the cytosol. Of course, not all of them are present in all cells, nor all of them are quantitatively important; furthermore, the expression level of many genes coding for enzymes using NAD^+^ and/or NADH as ligands exhibit diurnal rhythms on the day scale and/or respond to fasting or feeding [93]. However, the following three are very high flux enzymes, heavily influencing the cytosolic NADH/NAD^+^ ratio as a function of the concentration of the reactants. These are: (i) lactate dehydrogenase (LDH), (ii) glyceraldehyde-3-phosphate dehydrogenase (GAPDH), and (iii) cytosolic malate dehydrogenase (MDH1). To this list, the malate/aspartate shuttle should be added, mediating the transfer of reducing equivalents from the cytosol to mitochondria and vice versa [95]. All of the aforementioned reactions are connected under the following scheme: cells perform glycolysis during which NADH is formed by GAPDH, eventually yielding pyruvate; pyruvate may enter mitochondria under conditions supporting oxidative phosphorylation, otherwise becoming lactate and NAD^+^, the latter supporting GAPDH reaction; MDH1 may also support GAPDH by means of providing NAD^+^, especially in cells with mitochondrial dysfunction [96]; the malate/aspartate shuttle would be the means of transferring the reducing equivalent of NADH made by GAPDH to mitochondria, for the purpose of its oxidation in the electron transport chain. Evidently, none of the above would occur if mitochondrial dysfunction would lead to either a buildup of matrix reducing equivalents or disfavoring the malate/aspartate shuttle by means of decreasing the mitochondrial membrane potential or altering substrate gradients across the inner mitochondrial membrane. All these would eventually hinder the flow of reducing equivalents from the cytosol to the matrix through the shuttle. Thus, although the metabolic state does not correlate with the lysine succinylation extent of mitochondrial proteins [71], it may well do so with the cytosolic proteins, simply due to a decrease in cytosolic NAD^+^ availability, just as suggested in [72,73,74].

## 6. Glia in the Adult Human Brain Lack Succinyl-CoA and KGDHC Subunits

It is a textbook definition that under physiological conditions, the citric acid cycle operates in all cells harboring mitochondria; however, in 2015 we reported that glia in the adult human brain lack subunits of succinate-CoA ligase, the citric acid cycle enzyme responsible for performing the reversible conversion of succinyl-CoA and ADP (or GDP) to CoASH, succinate, and ATP (or GTP) [97]. Specifically, it was shown that immunoreactivity of subunit SUCLA2 of succinate-CoA ligase in surgical human cortical tissue samples was present exclusively in neurons [98], while subunit SUCLG2 immunoreactivity was confined to cells outlining the cerebral microvasculature [99], see Figure 2 and Figure 3, respectively. Succinate-CoA ligase is a heterodimeric enzyme, composed of an invariant α subunit encoded by SUCLG1 and a substrate-specific β subunit, encoded by either SUCLA2 or SUCLG2. This dimer combination results in either an ATP-forming (EC 6.2.1.5) or a GTP-forming SUCL (EC 6.2.1.4). Thus, if both SUCLA2 and SUCLG2 are missing, there cannot be succinate-CoA ligase activity. This would necessitate a bypass of succinate-CoA ligase in order to “close” the citric acid cycle in glial cells, and we proposed that the GABA shunt and/or enzymes participating in ketone body metabolism could substantiate this [99]. However, in 2020, we discovered that glia in the adult human brain also lack KGDHC-specific subunits [100], see Figure 4. This means that the ability of yielding succinyl-CoA within the mitochondrial matrix of glial cells is severely restricted, and if it happens, it would only be due to the GABA shunt [101] and/or enzymes catabolizing ketone bodies [102]. On the other hand, robust protein lysine succinylation was observed in the very same cells (Figure 5). The spatial resolution of immunohistochemistry for succinylated protein lysines is not sufficient for colocalization with mitochondrial markers; it is however, evident that lysine succinylation was labelling the cytosol of glia. The lack of succinate-CoA ligase and KGDHC subunits precluding the possibility of enzymatic activity, and thus formation of succinyl-CoA to an appreciable extent (GABA shunt, along with ketone body metabolism, does not encompass succinyl-CoA while it is on low-flux pathways), raises the question as to how there can be protein lysine succinylation in adult human glia. The answer probably lies in the expression profile of peroxisomal enzymes.

## 7. Glia in the Adult Human Brain Express Enzymes Supporting Peroxisomal Succinyl-CoA Production

The Allen Brain Atlas is an integrated spatio-temporal portal for exploring the human central nervous system [103]. It entails RNA-Seq data from 15,928 intact nuclei of middle temporal gyri derived from frozen human brain specimens collected from eight donors, aged 24–66 years. As shown in Figure 6, RNA-Seq data for enzymes participating in peroxisomal beta-, omega-3, and omega-6 fatty acid oxidation (indicated on the left) are depicted as a function of cell subtype. Glial cells are demarcated with a red dash (top right corner). It is immediately apparent that glial cells express many of the genes required for biochemical pathways that lead to succinyl-CoA production. It is therefore highly probable, that protein lysine succinylation observed in adult human glia stems from peroxisomal—and not mitochondrial- succinyl-CoA production. The importance of peroxisomes for glial protein lysine succinylation may add to the understanding of vital mechanisms other than myelination [104].

## 8. Conclusions

Proteins lysine succinylation depends on the provision of the “activated” molecule, succinyl-CoA. Although this is most likely true, it has not been unequivocally shown that lysine succinylation may also occur from succinate, using energy from another molecule. Thus, unless proven otherwise, it will be assumed that succinyl-CoA is critical and sufficient for this post-translational modification. Extramitochondrial protein lysine succinylation may be possible due to peroxisomal succinyl-CoA production. For this, (i) transport of succinyl-CoA outside peroxisomes (in the form of succinylcarnitine) and (ii) conversion of succinylcarnitine to succinyl-CoA by CPT are required. Therefore, extramitochondrial protein lysine succinylation could be dependent on SLC25A20 (the primary transporter of succinylcarnitine across the peroxisomal membrane) and CPT. Desuccinylation of cytosolic proteins is probably dependent on iso1 and iso4 isoforms of sirtuin SIRT5. Experimental data validating—or refuting—these claims are awaited.

## Figures and Tables

**Figure 1 ijms-22-06085-f001:**
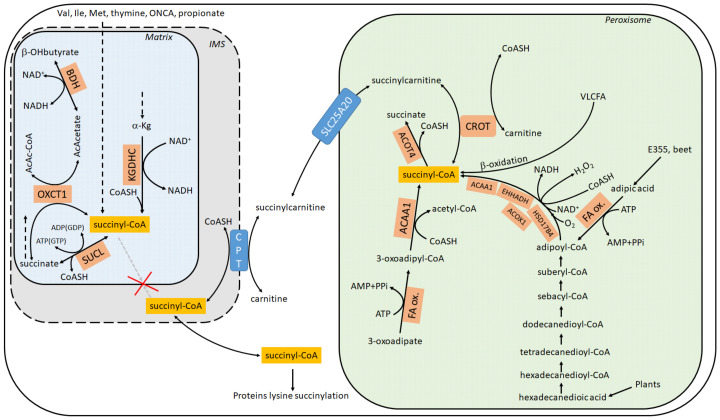
Metabolic pathways leading to succinyl-CoA, in cytosol, mitochondria and peroxisomes. ACAA1: acetyl-CoA acyltransferase 1; ACOT4: acyl-CoA thioesterase 4; ACOX1: acyl-CoA oxidase 1; a-Kg: alpha-ketoglutarate; BDH: beta-hydroxybutyrate dehydrogenase; b-OHbutyrate: beta-hydroxybutyrate; CoASH: coenzyme A; CPT: carnitine palmitoyltransferase (I); CROT: carnitine O-octanoyltransferase; E355: manufactured E number food additive; same as adipic acid; EHHADH: enoyl-CoA hydratase and 3-hydroxyacyl CoA dehydrogenase; FA ox.: fatty acid oxidation HSD17B4: hydroxysteroid 17-beta dehydrogenase 4; Ile: isoleucine; IMS: intermembrane space; KGDHC: a-ketoglutarate dehydrogenase; Met: methionine; ONCA: odd number chain fatty acid; OXCT1: 3-oxoacid CoA-transferase 1; SUCL: succinate-CoA ligase: Val: valine: VLCFA: very long chain fatty acid.

**Figure 2 ijms-22-06085-f002:**
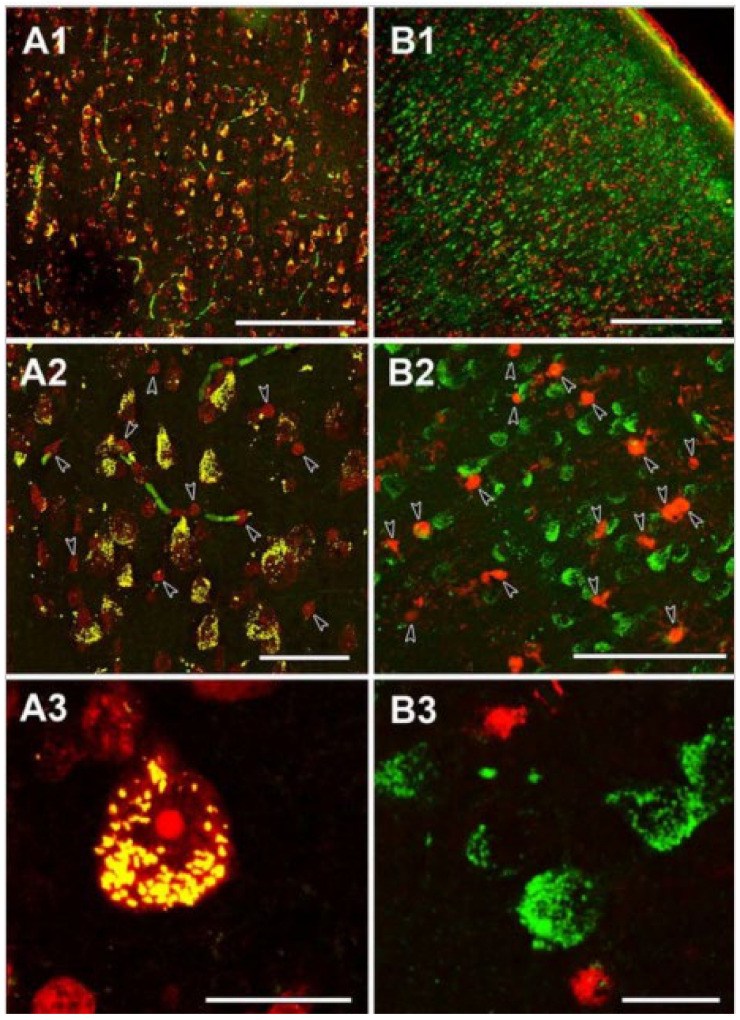
The neuronal localization of SUCLA2 immunoreactivity in the human temporal cortex. (**A1**): The distribution of SUCLA2-positive cells is similar to that of Nissl-labeled cells (red), both present throughout the temporal cortex. (**A2**): A higher magnification image reveals that essentially all large Nissl-labeled cells with irregular shape demonstrate SUCLA2 immunolabeling (yellow cells). In contrast, small, intensely labeled round-shape cells do not exhibit SUCLA2 immunoreactivity and remain red (arrowheads). Some blood vessels are labeled green because of the peroxidase activity of the red blood cells present in the vessels in the non-perfused human tissue. (**A3**): A high-magnification confocal microscopy image demonstrating the punctuate location of SUCLA2 immunoreactivity within the Nissl-labeled neurons whereas small glial cells do not exhibit SUCLA2 immunoreactivity. (**B1**): The distribution of the glial marker S100 (red) is different from that of SUCLA2-positive cells (green). (**B2**): A higher magnification image demonstrates lack of colocalization between SUCLA2 and S100. Glial cells labeled with S100 are indicated by arrowheads. (**B3**): High-magnification confocal picture shows that SUCLA2 and S100 are located in different cell types. Scale bars 200 μm for (**A1**), 50 μm for (**A2**), 20 μm for (**A3**), 500 μm for (**B1**), 50 μm for (**B2**), and 20 μm for (**B3**). (Adapted from [98], by permission).

**Figure 3 ijms-22-06085-f003:**
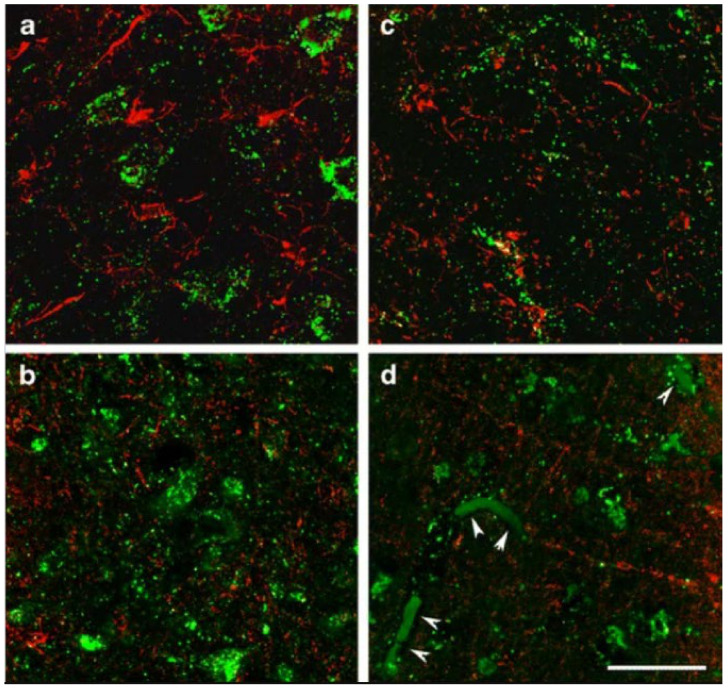
Immunohistochemistry of SUCLG2 and glial markers in the adult human brain. High magnification confocal images demonstrate that SUCLA2 immunoreactivity (green in (**a**,**b**)) is present in neurons but not in Iba1-positive microglia cells (red in (**a**)) and in oligodendroglial cells labelled with basic myelin protein (red in (**b**)). In turn, SUCLG2 immunoreactivity (green in (**c**,**d**)) does not co-localize with Iba1 (red microglia cells in (**c**)) and cells labelled with basic myelin protein (red oligodendroglial cells in (**d**)). The arrowheads in (**d**) point to red blood cells within vessels labelled non-specifically due to their internal peroxidase content. Scale bar = 20 μm. (Adapted from [99], by permission).

**Figure 4 ijms-22-06085-f004:**
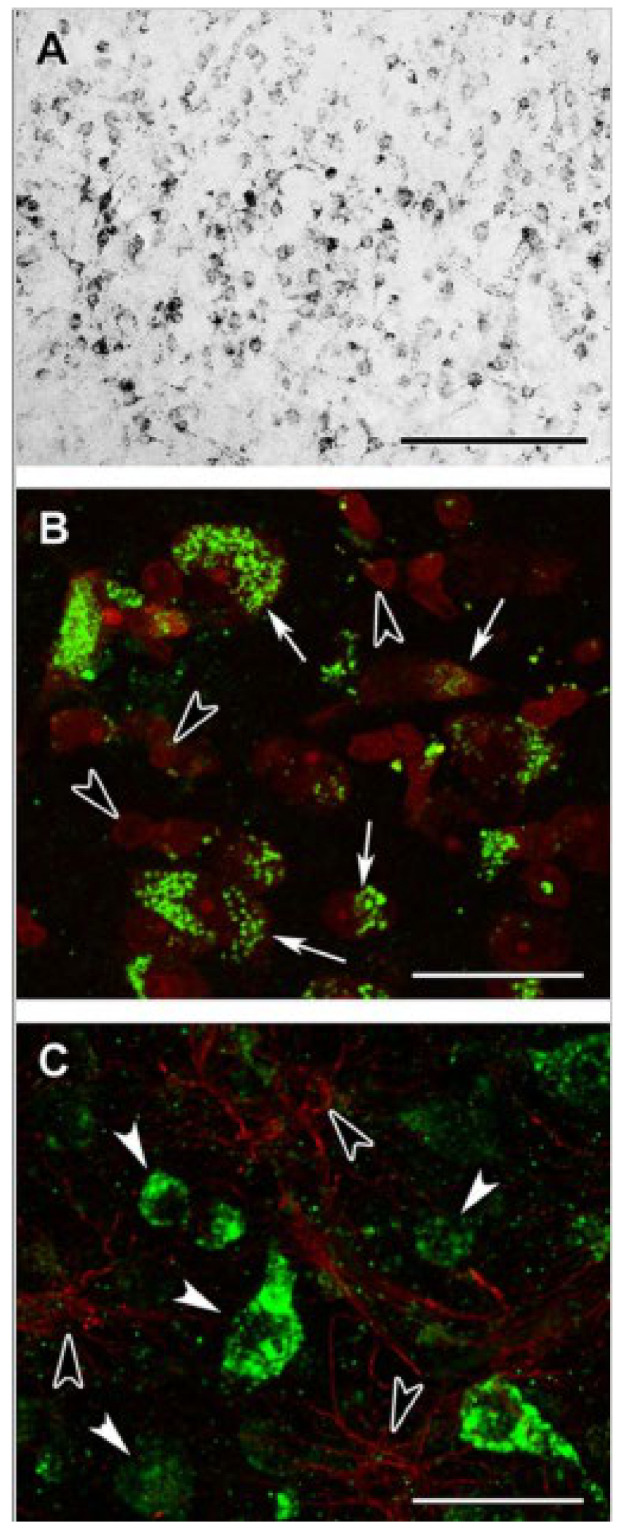
DLST (Dihydrolipoyllysine-residue succinyltransferase, E2 subunit of KGDGC) immunolabeling in the human cerebral cortex in relation to neuronal and glial markers. (**A**): DLST-immunoreactive (DLST-ir) cells are present in the cerebral cortex, with higher density in the deep layers. (**B**): DLST (green) and fluorescent Nissl staining (red) show that many cells are double-labeled in the cerebral cortex. Arrows point to double-labeled neurons, while black arrowheads point to single-labeled, DLST-immunonegative cells. Note the dot-like distribution of DLST immunolabeling in the double-labeled neurons. (**C**): A cerebral cortical section double-labeled with DLST (green) and the established astrocyte marker glial fibrillary acidic protein (GFAP; red) to show the lack of double-labeling, suggesting that GFAP-expressing astrocytes are devoid of DLST. White arrowheads point to single-labeled (DLST-ir) neurons; black arrowheads point to single-labeled (GFAP-ir) astrocytes. Scale bars = 300 µm for (**A**), and 30 µm for (**B**,**C**). (Adapted from [100] under the terms of the Creative Commons CC BY license).

**Figure 5 ijms-22-06085-f005:**
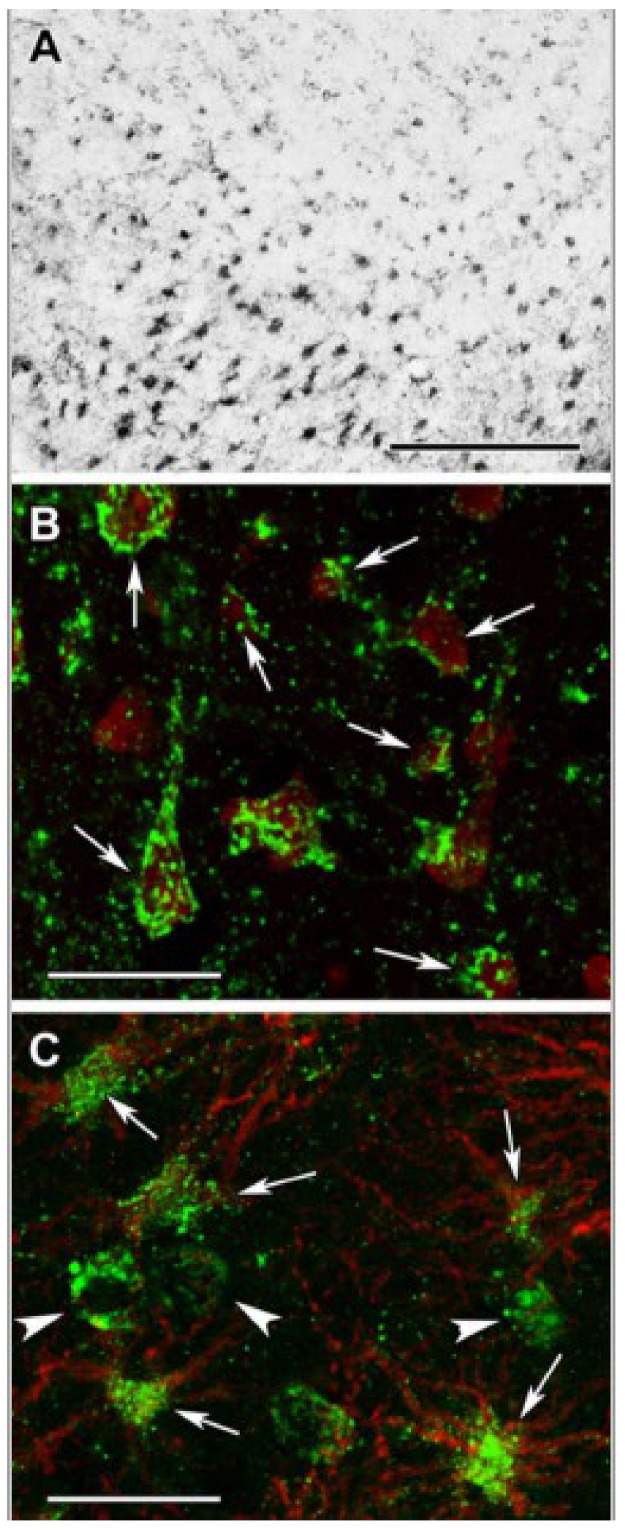
Succinyl-lysine immunolabeling in the human cerebral cortex in relation to neuronal and glial markers. (**A**): Succinyl-lysine-immunoreactive (SUCCLYS-ir) cells are present in the cerebral cortex, with higher density in the deep layers. (**B**): Succinyl-lysine (green) and fluorescent Nissl staining (red) show that essentially all cells (both neurons and glia) are double-labeled in the cerebral cortex. Arrows point to some of these double-labeled cells. Note the dot-like distribution of succinyl-lysine immunolabeling in the double-labeled cells. (**C**): A cerebral cortical section double-labeled with succinyl-lysine (green) and the established astrocyte marker glial fibrillary acidic protein (GFAP; red) to show the double-labeling of astrocytes. White arrowheads point to single-labeled (SUCCLYS-ir) neurons. Scale bars = 300 µm for (**A**) and 30 µm for (**B**,**C**). (Adapted from [100] under the terms of the Creative Commons CC BY license).

**Figure 6 ijms-22-06085-f006:**
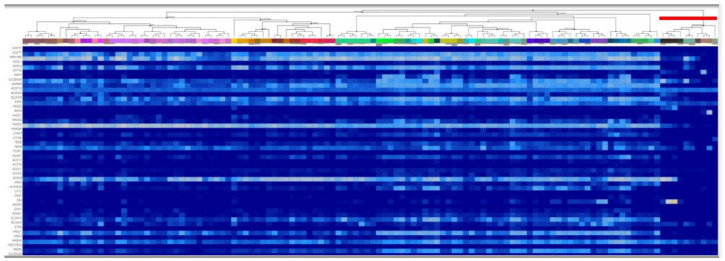
RNA-Seq data obtained from the Allen Brain Atlas, looking at the expression of genes (indicated on the left) participating in peroxisomal carnitine shuttle and fatty acid oxidation (beta-, omega-3, and omega-6), as a function of cell types. Glial cell types are indicated by the red dash in the top right corner. Image credit: Allen Institute: https://celltypes.brain-map.org/rnaseq/human (accessed on 25 May 2021).

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
