# Peer review of "The Mystery of Extramitochondrial Proteins Lysine Succinylation"

_ijms, 2021, doi:10.3390/ijms22116085_

Round 1

Reviewer 1 Report

The manuscript "The mystery of extramitochondrial proteins lysine succinylation", written by Christos Chinopoulos is a review describing possible biochemical pathways involved in succinyl-CoA production. The process of lysine succinylation is a posttranslational modification of proteins, present both, in mitochondria and in cytosol. While the processes in the mitochondria are well known, those present in the cytosol are not. The manuscript describes the possible pathways and enzymes, hypothetically involved in succinylation metabolism in the peroxisomes. In the second part of the manuscript hypothesis is further supported by data showing the presence of required enzymes for succinyl-CoA production in peroxisomes in glia cells which, at the same time, lack those in mitochondria.

The manuscript is oriented on proving the hypothesis of peroxysomal origin of cytosol succinylation presenting numerous biochemical reactions. In the Background I miss the description of the succinylation roles in general in the cell, its localization, scheme of the protein succynilation. There is no description of its role in the nucleus and pathways to these reactions.

The second paragraph could have a simple explanation of the role of succinate Co A ligase and description of reactions.

Sentences are sometimes too long and complex, with insertions which are not necessary. There is no description of all abbreviations when mentioned for the first time (i. e. KGDHC line 26, coASh line 47, SUCLA line 227). There is also no explanation of abbreviation in Figure 1. Figures 2-5 lack the title. The origin of the figures can be described at the end of the legend. There is no explanation of abbreviations (such as SUCLA2 and SUCLG2, DLST). Text on Figure 6 is too small to be readable. In fact, as immunostaining figures are not original but taken from another article, possibly they could be replaced with the data description: while certain proteins were detected in nerve cells, they are not present in glia cells.

There is no need of addressing to the reader – like see under Intraperoxisomal... (line 29, 68) or see figure 1 (line 120). Some sentences need to be reformulated (lines 87, 150,167...). There is no need of citing PMID number (line 240). In the fifth paragraph there is no need of SIRT1-4 description, as they are not the topic of the article. In Conclusion there is no explanation of CPS.

Minor:

line 8: no need of the sentence beginning with "Mindful..."

line 10: sentence reformulation

lines 14 and 16: better to avoid inserted sentences

line 58: sentence reformulation

line 79: if the data are cited specifically, it is better to mention the author

line 87: "in aggregate "  sentence reformulation

line 145, 150: sentence reformulation

line 244-248: the sentence is an indirect question, it does not need question mark.

Author Response

I thank Reviewer #1 for the comments.

In the Background I miss the description of the succinylation roles in general in the cell, its localization, scheme of the protein succynilation. There is no description of its role in the nucleus and pathways to these reactions.

> Response: It is my intention to provide an extremely focused mini-review on extramitochondrial proteins lysine succinylation, and keep it “undiluted” from generalizations regarding succinylations. From my perspective, I don’t wish to give to the reader the initial false impression “oh, another review capitalizing on the hot topic of succinylation”, I want to delve directly into the pertaining topic of extramitochondrial proteins lysine succinylation. On the other hand, minor text was added throughout the manuscript regarding the role in nucleus.

The second paragraph could have a simple explanation of the role of succinate Co A ligase and description of reactions.

> Response: I believe this is addressed by the text “ [7]. In either case, the the high-energy metabolite, succinyl-CoA, is required. It is a common misconception that succinyl-CoA is exclusively produced inside the mitochondrial matrix by -ketoglutarate dehyodrogenase complex (KGDHC) or reversal of succinate-CoA ligase (SUCL), or through the catabolism of valine, isoleucine, methionine, thymine, odd-number chain fatty acids (and perhaps propionate) and through the reaction catalyzed by 3-oxoacid CoA-transferase 1 (OXCT1), see figure 1.”, also shown in figure 1.

Sentences are sometimes too long and complex, with insertions which are not necessary.

> Response: The manuscript has been revised, and too long and complex sentences have been simplified.

There is no description of all abbreviations when mentioned for the first time (i. e. KGDHC line 26, coASh line 47, SUCLA line 227).

> Response: In the revised manuscript, abbreviations are described when mentioned for the first time.

 There is also no explanation of abbreviation in Figure 1.

> Response: In the revised manuscript, abbreviations are added in the legend of figure 1.

Figures 2-5 lack the title.

> Response: In the revised manuscript, titles were added in figures 2-5.

The origin of the figures can be described at the end of the legend.

> Response: In the revised manuscript, origin of the figures is described at the end of the legend.

 There is no explanation of abbreviations (such as SUCLA2 and SUCLG2, DLST).

> Response: In the revised manuscript, abbreviations are described when mentioned for the first time.

Text on Figure 6 is too small to be readable.

> Response: This is an image created automatically by the Allen Brain Atlas that I cannot modify. However, the image that I submitted to the journal can be expanded (as it is in png format) and there the letters are readable. I assume that this can happen through the browser, if viewing the manuscript online and not through a word processor or upon printing.

In fact, as immunostaining figures are not original but taken from another article, possibly they could be replaced with the data description: while certain proteins were detected in nerve cells, they are not present in glia cells.

> Response: I believe that the text relevant to the immunostaining figures do exactly that; also, in my opinion, it is helpful to include the original images, as they may help to convey the message “certain proteins were detected in nerve cells, they are not present in glia cells”.

There is no need of addressing to the reader – like see under Intraperoxisomal... (line 29, 68) or see figure 1 (line 120).

> Response: I believe that this is up to the policy of the journal; I have published papers in which there was indeed no need of addressing to the reader – like see under Intraperoxisomal, where other journals explicitly required to indicate where do the authors references such claims. I assume this will be settled in the proofs, if the current manuscript will reach this stage.

Some sentences need to be reformulated (lines 87, 150,167...).

> Response: The manuscript has been revised, and several sentences throughout the text were reformulated.

There is no need of citing PMID number (line 240).

> Response: In the revised manuscript, PMID numbers have been replaced with correct references.

In the fifth paragraph there is no need of SIRT1-4 description, as they are not the topic of the article.

> Response: I agree with Reviewer #1 that they are not the topic of the article, but because the main players are iso1 and iso4 of SIRT5 I wanted to be sure that the reader does not confuse iso1 and iso4 with SIRT1 and SIRT4, hence I included a description for all of them.

In Conclusion there is no explanation of CPS.

> Response: CPT was mistakenly named CPS, this has been corrected in the revised manuscript, including in figure 1; CPT is described in the text.

Minor:

line 8: no need of the sentence beginning with "Mindful..."

line 10: sentence reformulation

lines 14 and 16: better to avoid inserted sentences

line 58: sentence reformulation

line 79: if the data are cited specifically, it is better to mention the author

line 87: "in aggregate "  sentence reformulation

line 145, 150: sentence reformulation

line 244-248: the sentence is an indirect question, it does not need question mark.

> Response: Minor comments have been addressed in the revised manuscript. 

Reviewer 2 Report

This mini-review interestingly describes the potential extra-mitochondrial pathways underpinning lysine succinylation, a post-translational modification of proteins. In particular, pathways participating in peroxisomal fatty acid oxidation leading to succinyl-CoA production and the role of cytosolic SIRT5 iso1 and iso4 forms are presented and discussed. Finally, the discovery that glia in the adult human brain lacking subunits of both alpha- 
ketoglutarate dehydrogenase complex and succinate-CoA ligase -thus being unable to produce sucinyl-CoA in the matrix- and yet exhibit robust pancellular lysine succinylation, is highlighted. 

This Mini-Review is interesting, well written and referenced. It prepares the ground for new research aimed at investigating what authors here claim. 

I would suggest a minor revision of the English.

Author Response

I thank Reviewer #2 for the comments.

I would suggest a minor revision of the English.

> Response: The manuscript has been revised for English syntax and grammar.

Round 2

Reviewer 1 Report

The author of the manuscript "The mystery of extramitochondrial proteins lysine succinylation" responded to all of the comments.

There are still some minor comments:

It is common to indicate to the figure (or a chapter) by putting the number of the figure in the parentheses, instead of writing "see...." Figures should be cited with capital letters.

In lines 65 and 100 indirect speech sentences don't have a question mark.

line 127: thought

line 194: new sentence should begin with capital letter